

# Comment on "Short-cut transport path for Asian dust directly to the Arctic: a case Study" by Huang, Z., J. Huang, T., Hayasaka, S. Wang, T. Zhou and H. Jin (2015) in Environ. Res. Lett.

Keyvan Ranjbar[1], Norm T. O'Neill[1], Yasmin Aboel-Fetouh[1]

[1]Dépt. de géomatique appliquée, Centre d'Applications et de Recherches en Télédétection, Université de Sherbrooke, Sherbrooke, QC, Canada

*Correspondence to*: Keyvan Ranjbar (Keyvan.Ranjbar@usherbrooke.ca)

**Abstract**

The suggestion of Huang et al. (2015) on the climatological-scale transport of Asian dust to the Arctic appears to be an
important and worthwhile assertion. It is unfortunate that the authors undermined, to a certain degree, the quality of that assertion by a misinterpretation of the critical March 24, 2020 Arctic event (which was chosen by the authors to illustrate their generalized, climatological scale Arctic transport claim). They attempted to characterize that key event using AERONET/AEROCAN retrievals taken a day later and misinterpreted those largely cloud-dominated retrievals as being representative of Asian dust while apparently not recognizing that the coarse mode aerosol optical depth retrievals on the
previous day were actually coherent with their Arctic transport hypothesis.

**Introduction**

We recently came upon an interpretation in Huang et al. (2015) of Eureka AERONET/AEROCAN retrievals from the "PEARL" (actually Ridge Lab) instrument and from the Arctic High Spectral Resolution lidar (AHSRL) for data acquired in March of 2010 at the high-Arctic PEARL (Polar Environment Atmospheric Research Laboratory) complex. We note that the
authors of this comment are the mentors of the AERONET ("PEARL") instrument and longtime users of the Arctic High Spectral Resolution Lidar (AHSRL).

The suggestion of Huang et al. (2015) on the climatological-scale transport of Asian dust to the Arctic over a multi-year period, appears to be an important and worthwhile assertion. It is unfortunate that they undermined the quality of that assertion by a misinterpretation of the critical March 24, 2010 Arctic event (which was chosen by the authors to illustrate their generalized,
multi-year Arctic transport claim). They attempted to characterize that key event using AERONET/AEROCAN retrievals taken a day later and misinterpreted those retrievals while apparently not recognizing that the optically weak plume and the coarse mode aerosol optical depth (AOD) retrievals observed on the previous day were actually coherent with their HYSPLIT Arctic transport evidence.

**Dust and cloud events of March 24th and 25th 2010**

We believe that the event on the 2nd day (March 25, 2010) consisted of a complex but weak coarse mode AOD plume structure which was dominated by what was very likely a cloud intrusion after ~ 18:30 UT (c.f. the neighbourhood of the March 25th dashed vertical line in Figure 1). We can fairly confidently declare a cloud intrusion because of (i) the corresponding high

AHSRL depolarization ratio seen in Figure 1 and (ii) the strong variation of the coarse mode AOD which is much more typical of spatially inhomogeneous (hi-frequency) cloud than the low-frequency variation due to dust transported over large distances

(see O'Neill et al., 2016 for a similar interpretation of spatially inhomogeneous and homogeneous clouds and/or low altitude crystals over the nearby 0PAL site). The coarse mode AOD is a standard AERONET product which was available to the authors (see, for example, an illustrative comparison of PEARL coarse mode AOD and fine mode AODs with AHSRL-derived coarse mode and fine mode AODs in Saha et al., 2010).

The criticism of confusing dust and clouds is not unrelated to the fact that the authors neglected to consider potential problems

associated with the quality of the cloud screening algorithm. Their utilization of a significant March 25th drop in the value of the Angstrom exponent (AE) as an essentially qualitative indicator of the presence of coarse mode dust (an argument made in reference to their Figure 3), severely overestimated the optical depth of Asian dust (they also neglected to exploit the benefit provided by an analysis of the corresponding AHSRL profile). They should have limited their analysis to the March 24th, HYSPLIT-synchronized time period when the coarse mode AOD decreases by a slight amount (from a point just beyond the

stronger and broader values of the 7 km backscatter plume noted by the authors) by ~ 0.005 for Version 2 Level 1.0 coarse mode AOD retrievals (see Figure 1 for details). The backscatter coefficient profile of their Figure 2d (supported by their HYSPLIT-generated transport pathway of Figure 2i) suggests a dust plume arriving at Eureka on August 24 at an altitude close to the 7 km altitude of our Figure 1 AHSRL plume (a higher spatial- and colour-resolution version of their Figure 2d AHSRL profile): that 7 km plume is typical of the optically weak dust plumes observed over Eureka. The small (~ 0.005) decrease in

coarse mode AOD is the type of springtime variation that one expects for Asian dust (AboEl-Fetouh et al., 2020): not the March 25th, cloud-enhanced increase of ~ 0.05 (i.e. ~ 10 times the March 24th decrease).

This lack of optical coherence at the event level undermines their climatological scale claims of a preferred transport pathway of Asian dust into the Arctic. Their "probability-density-function (PDF)" computations of a preferred pathway to the Arctic are ostensibly convincing from a meteorological standpoint: however, the rigour of such affirmations can be called into

question if their chosen illustrations of dust events are suspect (if the synchronicity of coarse mode AOD and lidar profile variations is not investigated in detail and if the impact of potential cloud-screening problems is not properly examined).

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

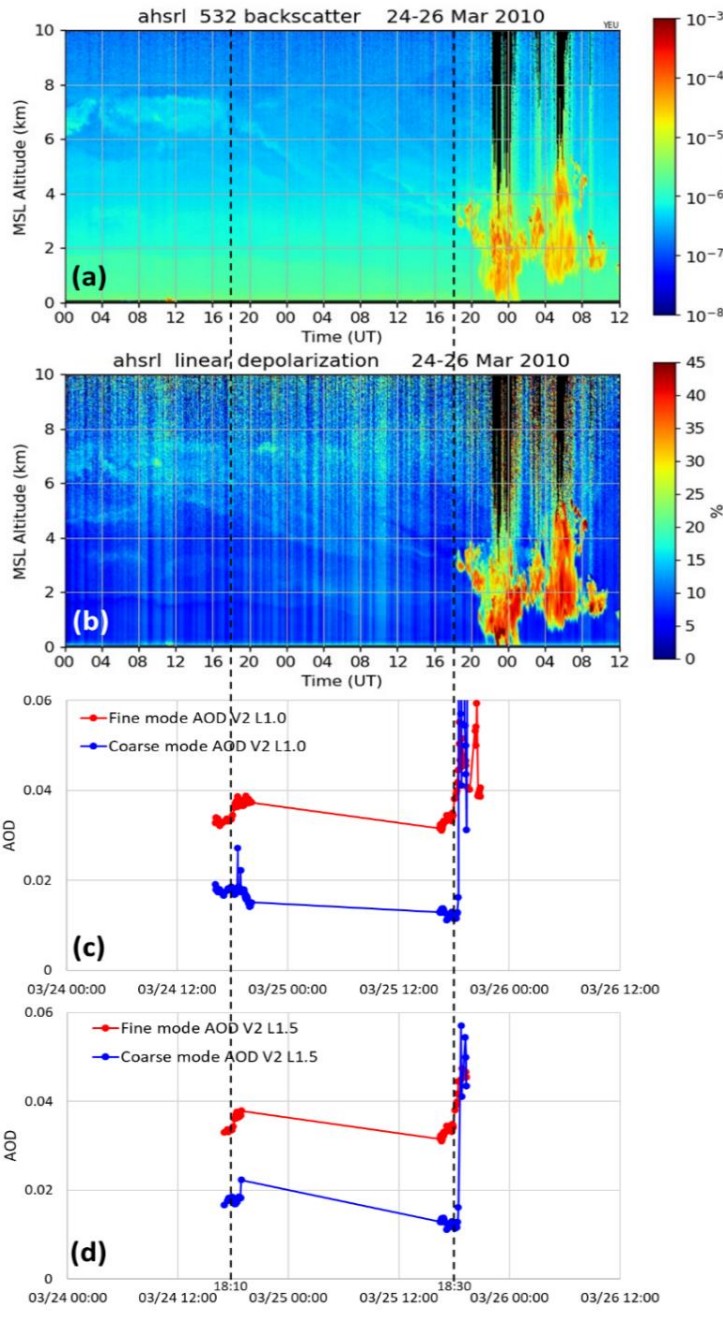






**Figure 1: (a) AHSRL backscatter coefficient (β) profile. (b) AHSRL linear depolarization (δ) profile. (c) Level 1.0 (non cloud-screened) fine mode and coarse mode AODs. (d) Level 1.5 (cloud-screened) fine mode and coarse mode AODs (a standard AERONET product). All times are UT. The March 24, 18:10 dashed vertical line indicates the beginning of the decrease in thickness of the coarse mode plume seen in the weak β profile and the moderate δ profile with an attendant weak decrease in the coarse mode**

**AOD (decrease ~ 0.005 from approximately 0.02 to 0.015 for the Version 2 Level 1.0 coarse mode AOD retrievals). The same decrease (but for different individual magnitudes of ~ 0.01 to 0.005) are obtained using Version 3 Level 1.0 retrievals (which were unavailable when the authors wrote their paper). The March 25, 18:30 dashed vertical line indicates the beginning of the rapid increase of the coarse mode AOD due to what is very likely cloud associated with the sharp increases in the β and δ profiles (cloud presence is typically associated with high frequency (rapid) coarse mode AOD increases). It should be emphasized that the V2 Level 1.5 (cloud-**

**screened) retrievals did not succeed in eliminating certain high frequency coarse mode AOD variations near the rapid rise at 18:30. V3 Level 1.5 retrievals did eliminate the coarse mode AOD in the region of that rapid rise (but failed to eliminate some thinner cloud of relatively low coarse mode AOD a few hours later).**