# Peer review of "Comment on "Short-cut transport path for Asian dust directly to the Arctic: a case Study" by Huang, Z., J. Huang, T., Hayasaka, S. Wang, T. Zhou and H. Jin (2015) in Environ. Res. Lett."

_Atmospheric Chemistry and Physics, 2021_

## Author Comment (AC1)

Response to Reviewer#1's comments on' Comment on "Short-cut transport path for Asian dust directly to the Arctic: a case Study" by Huang, Z., J. Huang, T., Hayasaka, S. Wang, T. Zhou and H. Jin (2015) in Environ. Res. Lett." '

**General Comment of the reviewer:**

Huang et al. 2015 uses ground-based lidar as the primary determination of the dust presence and presents ground-based sun photometer results as a verification of those measurements. The premise of the importance of the potential dust pathway from Asia to the Arctic region is maintained but the Arctic impact of the extremely weak dust layer provided in the example case remains in question. In general, the Reviewer is in agreement with the Authors comment with the misinterpretation of dust transport to the Arctic station, PEARL, on March 25, 2010.

A contextualizing comment that, we think, sums things up nicely

**Specific Comments:**

Line 11:

The event occurred on March 24, 2010, and not "2020" as stated.

Corrected

Line 26:

On March 24, 2010, the backscatter and depolarization signals appear extremely weak and approach background levels. Is it possible the lidar detects only the edge of the dust plume?

Yes, in the sense that it is detecting the narrower dust plume that follows on the heels of the broader 7-km plume. We reorganized that particular part of the text so that specific point was clearer

Line 30:

Avoid the wording that "We believe"; the comment should be based on evidence and scientific interpretation.

"We believe" was replaced by "We maintain"

Line 37:

State that these are the spectral deconvolution algorithm (SDA) AODs.

The paragraph where the AERONET coarse mode AOD product is discussed was re-organized to include the SDA acronym expansion as well as the citation to O'Neill et al., 2003 (while actually decreasing the amount of text in parentheses; the paragraph was more cumbersome than it needed to be).

Lines 52-56:

Optical depth changes suggest an extremely weak dust plume, 0.005. Do the Author's maintain that this extremely small change in AOD (apparently corresponding to dust) has a significant impact on the Arctic region? For example, lidar does not indicate a descending layer near the

ground for deposition of dust in the region of PEARL. While Huang et al., 2015, indicates that "dust from 25.2% of Asian dust events generated during this period has potentially been transported directly to the Arctic," their example period (and verified by this current manuscript) shows an extremely small amount of dust reaching the Arctic likely due to deposition resulting from dynamic meteorological processes, precipitation removal, and/or uptake via cloud condensation nuclei for cloud ice crystals. Further elaboration is needed here in regard to these issues.

While this is a relevant issue, it is outside the scope of a short comment (on the optical-physics of a particular dust event), to have us shoulder the responsibility for explaining the deposition impact of Asian dust if its optical depth is significantly smaller than the illustration of Huang et al. We would, nonetheless, point the reviewer to Zwaaftink et al. (2016) whose major (Flexpart-driven) conclusion is the overwhelming importance of local (drainage-basin) dust (compared with long-range transport dust) to deposition : "*This leads to dominant contributions of local dust sources to total surface dust concentrations (~85%) and dust deposition (~90%) in the Arctic region.*"

Figure 1:

The sun photometer is viewing the sun at approximately 78 degrees solar zenith angle during these measurements at PEARL, which is significantly different than the zenith viewing angle of the HSRL. Do the Authors have further evidence of homogeneous spatial distribution of the dust plume and clouds to perform a more direct comparison with sun photometer measurements? If not, then this uncertainty must be clearly stated and discussed in context of these challenges in the Arctic environment. Further, please state the uncertainty of the fine and coarse mode AOD in the caption or text.

We have found, over the years (mostly for high-Arctic, low-SZA events), consistently strong correlations between photometrically-derived fine and coarse mode AODs (with inter-sample resolutions of >~ minutes at bin resolutions ~ minutes) and lidar fine and coarse mode AODs (derived from profiles of contiguous-bin resolutions of 10s of seconds) that are resampled to the photometric sampling bins. We have demonstrated this strong correlation in numerous papers (see, for e.g., O'Neill et al., 2004, O'Neill et al., 2008, Saha et al., 2010, O'Neill et al., 2012, Cottle et al., 2013, Baibakov et al., 2015, O'Neill et al., 2016, Ranjbar et al., 2019).

There is no doubt that a correlation degradation between the photometric and lidar fine and coarse mode AODs occurs with, notably, an increasing shift between their temporal patterns (such as, for example, the temporal shift attributed to the intrusion of a rapidly moving coarse mode cumulus cloud) but this is tempered by the actual temporal/spatial frequency of the event: fine and coarse mode aerosol events are generally of significantly lower temporal/spatial frequency than coarse mode clouds (the most extreme temporal shift occurring for clouds; but we do not analyze the optical properties of clouds). We have, in the past, carried out various checks on the degradation of the correlation with different retrieval parameters (see, for example, Baibakov et al., 2015): however, in the end, we appeal to the long-standing empirical finding that photometric and lidar fine and coarse mode AODs are generally always well correlated.

**References**

Baibakov, K., O'Neill, N. T., Ivanescu, L., Duck, T. J., Perro, C., Herber, A., Schulz, K.-H., and Schrems, O.: Synchronous polar winter starphotometry and lidar measurements at a High Arctic station, Atmos. Meas. Tech., 8, 3789-3809, doi:10.5194/amt-8-3789-2015, 2015.

Cottle, P., K. Strawbridge, I. McKendry, O'Neill, N., Saha, A., A pervasive and persistent Asian dust event over North America during spring 2010 : lidar and sunphotometer observations, ACP, 13, 4515-4527, 2013.

Groot Zwaaftink, C. D., H. Grythe, H. Skov, and A. Stohl (2016), Substantial contribution of northern high-latitude sources to mineral dust in the Arctic, J. Geophys. Res. Atmos., 121, 13,678–13,697, doi:10.1002/2016JD025482.

O'Neill, N. T., K. Baibakov, S. Hesaraki, L. Ivanescu, R. V. Martin, C. Perro, J. P. Chaubey, A. Herber, and T. J. Duck. "Temporal and spectral cloud screening of polar winter aerosol optical depth (AOD): impact of homogeneous and inhomogeneous clouds and crystal layers on climatological-scale AODs." ACP, 16, no. 19, 12753-12765, 2016.

O'Neill, N. T., Perro, C., Saha, A., Lesin, G., Duck, T., Eloranta, E., Hoffman, M. L. Karumudi, A., Ritter, C., A. Bourassa, I. Aboud, S. Carn, V. Savastiouk, Impact of Sarychev sulphate aerosols over the Arctic, Jour. Geophys. Res., VOL. 117, D04203, doi:10.1029/2011JD016838, 2012.

O'Neill, N. T., O. Pancrati, K. Baibakov, E Eloranta, R. L. Batchelor, J. Freemantle, L. J. B. McArthur, K. Strong, and R. Lindenmaier, Occurrence of weak, sub-micron, tropospheric aerosol events at high Arctic latitudes, Geophys. Res. Lett., 35, L14814, doi:10.1029/2008GL033733, 2008.

O'Neill, N. T., Strawbridge, K. B., Thulasiraman, S., Zhang, J. , Royer, A. , Freemantle, J., Optical coherency of Sunphotometry, sky radiometry and lidar measurements during the early phase of Pacific2001, accepted for publication in Atmospheric Environment, 2004

Ranjbar K, O'Neill NT, Lutsch E, McCullough EM, AboEl-Fetouh Y, Xian P, Strong K, Fioletov VE, Lesins G, Abboud I. Extreme smoke event over the high Arctic. Atmospheric Environment, 218, 117002 2019

Saha, A., N. T. O'Neill, E Eloranta, R. S. Stone, T. F. Eck, S. Zidane, D. Daou, A. Lupu, G. Lesins, M. Shiobara, L. J. B. McArthur, Pan-Arctic sunphotometry during the ARCTAS-A campaign, April 2008, Geophys. Res. Lett., Vol. 37, L05803, doi:10.1029/2009GL041375, 2010.